# Intra-Abdominal Nocardiosis—Case Report and Review of the Literature

**DOI:** 10.3390/jcm9072141

**Published:** 2020-07-07

**Authors:** Lucas Tramèr, Kirsten D. Mertz, Rolf Huegli, Vladimira Hinic, Lorenz Jost, Felix Burkhalter, Sebastian Wirz, Philip E. Tarr

**Affiliations:** 1University Department of Medicine, Kantonsspital Baselland, University of Basel, 4101 Bruderholz, Switzerland; lucas.tramer@ksbl.ch (L.T.); lorenz.jost@ksbl.ch (L.J.); felix.burkhalter@ksbl.ch (F.B.); sebastian.wirz@ksbl.ch (S.W.); 2Cantonal Institute of Pathology, 4410 Liestal, Switzerland; kirsten.mertz@ksbl.ch; 3Department of Radiology and Nuclear Medicine, Kantonsspital Baselland, 4101 Bruderholz, Switzerland; rolf.huegli@ksbl.ch; 4Clinical Bacteriology and Mycology, Laboratory Medicine, University Hospital Basel, 4031 Basel, Switzerland; vladimira.hinic@usb.ch; 5Oncology Service, Kantonsspital Baselland, 4101 Bruderholz, Switzerland; 6Nephrology Service, Kantonsspital Baselland, 4101 Bruderholz, Switzerland; 7Infectious Diseases Service, Kantonsspital Baselland, 4101 Bruderholz, Switzerland

**Keywords:** nocardiosis, *Nocardia paucivorans*, infection, abdominal, retroperitoneal, abscess, immunosuppression, malignancy

## Abstract

Nocardiosis is primarily an opportunistic infection affecting immunosuppressed individuals, in whom it most commonly presents as pulmonary infection and sometimes cerebral abscesses. Isolated abdominal or retroperitoneal nocardiosis is rare. Here, we report the second case, to our knowledge, of isolated abdominal nocardiosis due to *Nocardia paucivorans* and provide a comprehensive review of intra-abdominal nocardiosis. The acquisition of abdominal nocardiosis is believed to occur via hematogenous spreading after pulmonary or percutaneous inoculation or possibly via direct abdominal inoculation. Cases of *Nocardia* peritonitis have been reported in patients on peritoneal dialysis. Accurate diagnosis of abdominal nocardiosis requires histological and/or microbiological examination of appropriate, radiologically or surgically obtained biopsy specimens. Malignancy may initially be suspected when the patient presents with an abdominal mass. Successful therapy usually includes either percutaneous or surgical abscess drainage plus prolonged combination antimicrobial therapy.

## 1. Introduction

*Nocardia* are gram-positive, aerobic, filamentous bacteria that most often cause pulmonary infection in immunosuppressed hosts, while isolated abdominal or retroperitoneal nocardiosis is rare [1]. In extrapulmonary nocardiosis, malignancy is often suspected due to the presence of abdominal or cerebral masses [2,3,4,5,6], which can lead to misdiagnosis. Additionally, accurate diagnosis may be challenging due to slow growth of nocardia species in culture [7,8].

Here, we report the second case, to our knowledge, of isolated abdominal nocardiosis due to *Nocardia paucivorans*. We also provide a comprehensive review of the literature, summarizing the information on 39 previously published cases of intra-abdominal and retroperitoneal nocardiosis and 12 cases of peritoneal dialysis-related nocardia peritonitis [2,5,6,8,9,10,11,12,13,14,15,16,17,18,19,20,21,22,23,24,25,26,27,28,29,30,31,32,33,34,35,36,37,38,39,40,41,42,43,44,45,46,47,48,49,50,51,52,53]. Finally, we will review published cases of infection with *Nocardia paucivorans*, a rare nocardial species.

## 2. Illustrative Case Report

An 81-year-old man with diffuse large B-cell lymphoma presented in January 2019 with abdominal pain and night sweats. In 2009, he initially presented with abdominal pain, weight loss, night sweats, and fever. Abdominal CT showed a mesenteric mass adjacent to the ileocecal area. Histopathological examination of a surgically obtained biopsy specimen showed the presence of CD20- and CD3-positive lymphoma cells, a perifocal lymphohistocytiary inflammatory infiltrate as well as necrotic lymphoma tissue. A bone marrow biopsy was free of lymphoma cells. Weekly rituximab was administered for 4 weeks, in combination with vincristin and bleomycin for the first 3 weeks. Rituximab plus cyclophosphamide, hydroxydaunorubicin, oncovin and prednisolone (R-CHOP) was administered for a total of 6 cycles until full remission was achieved in February 2010. After a successfully treated low-grade nodal relapse in 2017, the patient remained asymptomatic until February 2018 when persistent hypogammaglobulinemia was first noted and intravenous immune globulin was given at 4-week intervals. Since the last relapse in 2017, the patient had received no immunosuppressive therapy.

In January 2019, the patient presented with generalized weakness, night sweats, and intermittent abdominal pain for the past 3 weeks, as well as fever and diarrhea for 1 week. On physical examination, the abdomen was soft and without tenderness. CRP (197 mg/L) and the white blood cell count (18,100/mm^3^) were elevated. A chest X-ray showed no signs of pulmonary infection, and on CT scan of the head there were no signs of cerebral nocardiosis. Abdominal CT showed a heterogeneous mass in the right adrenal region (Figure 1), with FDG-PET-CT showing locally increased FDG uptake and central necrosis. A biopsy specimen, radiologically obtained with CT-guidance revealed numerous granulocytes and histiocytes but no signs of malignancy, suggesting a retroperitoneal abscess.

A pigtail catheter was placed under CT-guidance for drainage of the abscess (Figure 1) and empiric antibiotic therapy with intravenous amoxicillin/clavulanic acid was initiated. Gram stain of the drainage fluid showed filamentous bacteria, consistent with *Nocardia species* (Figure 2). Culture grew *Nocardia paucivorans*, susceptible in vitro to ceftriaxone, imipenem, amikacin, trimethoprim-sulfamethoxazole (TMP-SMX), ciprofloxacin and minocycline and resistant to amoxicillin/clavulanic acid. *Nocardia paucivorans* was identified by matrix-assisted laser desorption ionization time-of-flight mass spectrometry (MALDI-TOF MS). The mass spectra were acquired using the Microflex LT system and analyzed with the MALDI Biotyper Compass 4.1 software (Bruker Daltonics, Bremen, Germany). The identification was confirmed by partial 16S rRNA gene sequencing as described previously [54]. Antimicrobial susceptibility testing was performed with Liofilchem^®^ MIC Test Strips (Liofilchem, Roseto degli Abruzzi, Italy) on Mueller Hinton agar +5% horse blood +20 mg/l ß-NAD (MHF; bioMérieux, Marcy-l’Étoile, France), and the minimal inhibitory concentrations (MICs) were interpreted according to CLSI guidelines [55]. PCR and culture were negative for mycobacteria. The route of acquisition of *Nocardia paucivorans* in this patient could not be identified.

Antibiotic therapy was changed to intravenous trimethoprim-sulfamethoxazole and amikacin, and the patient clinically improved. The trough-level of amikacin was monitored and kept in the recommended range. However, the patient developed acute renal failure (eGFR decrease from 50 to 14 mL/min) and metabolic acidosis. This prompted the change of antimicrobial treatment to ceftriaxone and minocycline, and renal function recovered.

Due to considerable residual abscess despite external drainage, surgical abscess drainage was performed and the patient further improved. He was discharged on oral minocycline and ciprofloxacin. On follow-up, six weeks postoperatively, he reported minimal abdominal pain, tolerated antibiotic therapy well, and blood inflammatory markers were normal. Antibiotic therapy was continued for a total of 6 months. On most recent follow-up, 6 months after antibiotics were discontinued, the patient was well, and abdominal ultrasound was unremarkable.

## 3. Methods

We reviewed the existing literature on abdominal nocardiosis and infections due to *Nocardia paucivorans*. To identify patients in the published literature, several PubMed searches were performed. Search items included “nocardiosis”, “nocardia”, “paucivorans”, “abdominal”, “retroperitoneal”, “CAPD”, “peritonitis” and “infection”. Additional cases were identified through bibliographic review of retrieved publications. Abdominal nocardiosis was defined as clinical or radiological signs compatible with nocardiosis plus isolation of *Nocardia species* in at least one abdominal or retroperitoneal specimen regardless of the infection’s origin and extent of dissemination. *Nocardia* peritonitis was defined as clinical signs of peritonitis plus isolation of *Nocardia species* in ascites or peritoneal dialysate. *N. paucivorans* infection was defined as clinical or radiological signs compatible with nocardiosis plus isolation of *N. paucivorans* in an appropriate specimen, with clinical or radiological signs of infection. We limited our review to English and German language publications since 1966. The initial searches led to 93 publications being retrieved, of which 29 were excluded (26 articles published prior to 1966, 2 publications in Japanese, 1 publication in Spanish). Therefore, the present review includes 64 publications, including 49 cases of *N. paucivorans* infection, 39 cases of abdominal or retroperitoneal nocardiosis, 12 cases of nocardial peritonitis, plus our own patient with abdominal *N. paucivorans* infection.

## 4. Literature Review and Discussion

In this review of the literature, we will first discuss the presumed route of acquisition of nocardiosis based on previous case reports and reviews. Second, we will discuss previously reported infections with *N. paucivorans*. Third, we will concentrate on the characteristics of abdominal nocardiosis and *Nocardia* peritonitis.

### 4.1. Route of Nocardia acquisition

*Nocardia* are ubiquitously present in soil, water and decaying vegetation [56]. Infection is believed to occur mostly through inhalation or percutaneously (i.e., via local trauma, skin cuts or scrapes); intestinal inoculation has at least been suggested to occur [7]. In immunocompetent individuals (1/3 of cases), nocardiosis tends to remain localized to the site of inoculation [7]. With immunosuppression, disseminated nocardiosis may occur, clinically most often presenting as brain abscesses [7,56,57]. Presumably, respiratory acquisition of *Nocardia* in immunosuppressed patients is followed by hematologic dissemination, eventually allowing infection at distant sites [22,38]. Additionally, direct peritoneal inoculation has been reported in 12 cases of *Nocardia* peritonitis in patients on peritoneal dialysis [8,43,44,45,46,47,48,49,50,51,52,53].

### 4.2. Infection Due to Nocardia Paucivorans 

N. paucivorans was first isolated in sputum in 2000 [58]. To our knowledge, 49 cases of N. paucivorans infection have been published [2,3,4,56,58,59,60,61,62,63,64,65,66,67,68], and we summarize these in Table 1. The majority (33 cases) were reported by Gray et al. from Queensland, Australia, and were identified retrospectively via 16S ribosomal DNA sequencing of archived biopsy specimens [63]. Unfortunately, the paper by Gray et al. lacks details regarding therapy and outcome of most patients, limiting the information to be gained from these cases. Additionally, none of the 49 previously reported cases included detailed information on the presumed route of acquisition of N. paucivorans infection [2,3,4,56,58,59,60,61,62,63,64,65,66,67,68].

In 18 of 49 cases (37%), no information was provided regarding the patients’ immune status. Of the remaining 31 cases, 11 patients (35%) were reported to be immunocompetent while 20 patients (65%) were overtly immunocompromised or had an underlying clinical condition. Thus, nocardiosis with N. paucivorans involves immunocompetent and immunocompromised patients, similar to infection with other Nocardia species [1,7].

Microbiologically documented lung involvement was reported in 29 of 49 cases (59%) of human N. paucivorans infection, while 19 cases (39%) involved the central nervous system (mainly brain abscesses). Additional cases include skin and muscle abscesses, bacteremia and endocarditis [3,4,56,58,59,60,61,62,63,65,66,68]. There is one case of abdominal/retroperitoneal N. paucivorans infection in the literature, a 63-year old lung transplant recipient with a renal abscess published by Roy in 2018 [2]. Our patient reported in Section 2 thus represents the second published case of intra-abdominal/retroperitoneal infection with N. paucivorans. N. paucivorans is typically susceptible in vitro to most commonly employed anti-nocardial agents [56] with TMP-SMX being used most frequently.

### 4.3. Abdominal/Retroperitoneal Infection Due to Nocardia Species 

There are 39 cases of abdominal/retroperitoneal nocardiosis reported in the English and German language literature since 1966 [2,5,6,9,10,11,12,13,14,15,16,17,18,19,20,21,22,23,24,25,26,27,28,29,30,31,32,33,34,35,36,37,38,39,40,41,42] (Table 2 and Table 3). In 16 of 39 cases (41%), concurrent involvement of the lungs and/or brain was recorded (Table 3). Isolated abdominal nocardiosis with no lung or brain involvement was reported in 23 previous cases (59%) (Table 2).

*N. asteroides* and *N. farcinica* (10 cases each) represent the most commonly isolated species. The route of *Nocardia* acquisition leading to abdominal or retroperitoneal infection remains unclear in most case reports. Some authors of published abdominal nocardiosis cases have speculated on the route of acquisition, which may have included inhalation of infected dust followed by hematogenous spread of *Nocardia species* (Table 2, case 23) [26], and acquisition via infected pet dogs with skin lesions due to distemper virus (Table 2, case 22) [25]. While direct intestinal inoculation has been suggested to occur [7], involvement of the intestines (jejenum, ileum or colon) was observed in only 6 cases (15%), suggesting that hematogenous spread from an extraabdominal inoculation site seems the more likely scenario in most cases.

Patients with impaired cell-mediated immunity such as solid organ transplant recipients and patients with advanced HIV infection may be at particular risk for contracting disseminated nocardiosis [7,57]. Risk factors for abdominal nocardiosis remain a matter of speculation. One patient (Table 2, case 14) had colon cancer with a resulting bowel perforation and abscess at this site [42]. N. veterana was cultured from the abscess, and the authors speculated that local tissue injury in the setting of the malignancy may have allowed the organism to establish infection here. In a patient with AIDS and renal nocardiosis (Table 2, case 13), the authors suggested that local kidney damage caused by urolithiasis with subsequent hydonephrosis might have predisposed the patient to localized nocardiosis of the kidney [18].

As in our patient reported in Section 2, isolation of Nocardia species in the renal or adrenal region was the most common infection site amongst previous cases of abdominal or retroperitoneal nocardiosis (15 cases, 38%). Further presentations include abscesses of the liver (7 cases, 18%), spleen (3 cases, 8%) and ovaries (2 cases, 5%). Two thirds of the patients (26 cases, 66%) had a least one immunocomprimising condition, in line with the literature on Nocardia species in general [1,7]. The mortality rate was high (36%; 14 of 39 cases). Only 3 patients who died were reported to be immunocompetent. Unsuprisingly, mortality was higher in patients with disseminated nocardiosis than in those with isolated abdominal or retroperitoneal disease. Overall, abdominal/retroperitoneal nocardiosis seems to affect primarily adults, with only 4 cases (10%) being diagnosed under the age of 18 years.

Accurate diagnosis of abdominal nocardiosis may be challenging due to the non-specific clinical and radiological findings. Additionally, Nocardia are often not visualized on initial stains, delaying clinical diagnosis and effective therapy [2]. Due to the slow growth of Nocardia species, adequate culture incubation of at least 5 days should be considered [2,8]. Abdominal nocardiosis is often misdiagnosed as tuberculosis [25,34] or abdominal malignancy such as adrenal [5], ovarian [19], thyroid [31] or renal carcinoma [2,39]. Misdiagnosis can lead to erroneous therapy including antituberculous treatment or even extensive surgery [19,25]. Suspected to have ovarian carcinoma, one published patient (Table 2, case 15) underwent total hysterectomy, salpingo-oophorectomy and local lymph node dissection before being diagnosed with abdominal nocardiosis [19].

Successful therapy typically included either percutaneous or surgical abscess drainage plus prolonged antimicrobial therapy ranging from 3 to 12 months. As for other forms of nocardiosis, effective antibiotics for abdominal Nocardia infections include trimethoprim-sulfamethoxazole (TMP-SMX), third-generation cephalosporins, extended-spectrum quinolones, carbapenemes, minocycline and linezolid [2]. In previous published cases of abdominal and retroperitoneal nocardiosis, TMP/SMX was the most commonly used antimicrobial agent (Table 2 and Table 3). Given the typically prolonged antimicrobial therapy and the frequent use of combination therapy, kidney and liver function as well as potential pharmacological interactions should be closely monitored as demonstrated by our case report (Section 2).

### 4.4. Peritoneal Dialysis-Related Nocardia Peritonitis

*Nocardia species* is a rare cause of peritonitis in patients on continuous ambulatory peritoneal dialysis (CAPD), with 12 published cases in the literature, which we summarize in Table 4 [8,43,44,45,46,47,48,49,50,51,52,53]. Peritonitis is an important complication leading to technique failure, hospitalization and even mortality in CAPD patients [53]. Regular fluid exchange via the permanent percutaneous catheter exposes CAPD patients (but not patients on hemodialysis) to an increased risk for bacterial peritoneal contamination, which occasionally may include opportunistic pathogens such as *Nocardia*.

Identifying and treating Nocardia peritonitis may be difficult for several reasons. First, the clinical presentation of Nocardia peritonitis typically is no different from ordinary bacterial peritonitis [47]. Second, Nocardia may tend to form clumps in liquid environments, dispersing non-uniformly in peritoneal fluid, which may explain why Nocardia peritonitis can be difficult to identify [45]. Third, Nocardia grow slowly in culture, and the infection often presents as a “culture-negative” peritonitis, unresponsive to empirical antibiotic treatment. Extending the incubation time of the dialysate to a minimum of 10 days should be considered in cases of culture-negative and antibiotic-non-responsive peritonitis in CAPD-patients [8]. The finding of slow growing gram-positive rods in dialysate culture should raise the suspicion of Nocardia peritonitis [49]. Trimethoprim-sulfamethoxazole (TMP/SMX), often in combination with other antimicrobial agents, appears to be the treatment of choice [50,51]. Intraperitoneal antibiotic treatment is the preferred route of administration. In 5 of the 12 published CAPD patients with Nocardia peritonitis, intraperitoneal antibiotic administration was successful without removal of the catheter and therefore without technique failure. In the remaining cases, due to initial treatment failure, the PD-catheter was removed, and the patient switched to maintenance hemodialysis, and antibiotic treatment was given orally or intravenously until full resolution of the infection [48,51].

## 5. Conclusions

Human nocardiosis is a rare opportunistic disease affecting primarily immunosuppressed adults. However, one third of patients have no immunocompromising condition [1,7,63]. The acquisition route of abdominal nocardiosis typically is unclear. Hematologic dissemination after pulmonary or percutaneous inoculation and direct abdominal inoculation have been suggested to occur [7,22,38]. In addition, 12 cases of direct peritoneal inoculation in patients with continuous ambulatory peritoneal dialysis have been reported [8,43,44,45,46,47,48,49,50,51,52,53]. In abdominal nocardiosis, malignancy is often suspected due to the presence of an abdominal mass [2,3,4,5,6]. In our patient, an FDG-avid mass on FDG-PET led to suspicion of malignancy recurrence. Thus, biopsy of these masses and appropriate microbiological tests (including modified acid-fast staining, PCR, and culture) is crucial. Because of slow growth of *Nocardia species*, cultures should be incubated for at least 10 days to avoid false-negative results [8]. *N. asteroides* and *N. farcinica* are the most commonly isolated species in abdominal nocardiosis. *N. paucivorans* is a rare species leading to abdominal nocardiosis with only one previous case report, our patient thus representing the second published case. As in our patient, isolation of *Nocardia* species in the renal or adrenal region was the most common infection site amongst previous cases of abdominal or retroperitoneal nocardiosis. To date, there are no therapy guidelines for abdominal nocardiosis. In reported cases, successful therapy usually included either percutaneous or surgical abscess drainage plus prolonged combination antimicrobial therapy including TMP/SMX ranging from 3 to 12 months, as recommended for other forms of nocardiosis [69]. As demonstrated by our case report, liver and renal function as well as potential pharmacological interactions should be closely monitored during antimicrobial therapy.

## Figures and Tables

**Figure 1 jcm-09-02141-f001:**
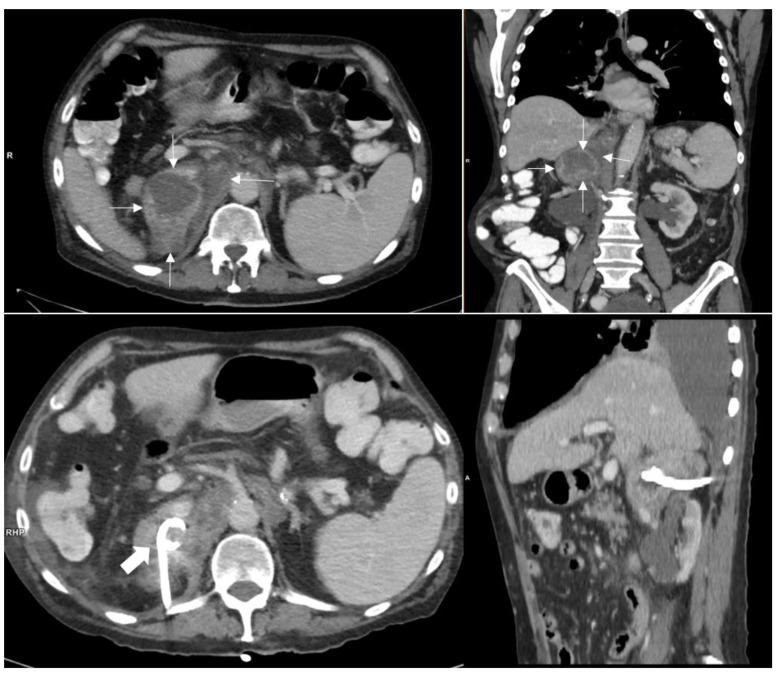
**Upper row**: 11 × 6 cm retroperitoneal mass in the right adrenal loge (thin white arrows) with contact to diaphragm, liver and retroperitoneal vessels. The pararenal space is not involved. The mass shows an inhomogeneous enhancement. The right adrenal gland itself cannot be distinguished. **Lower row**: Mass after puncture and CT guided drainage with a pigtail catheter (thick white arrow).

**Figure 2 jcm-09-02141-f002:**
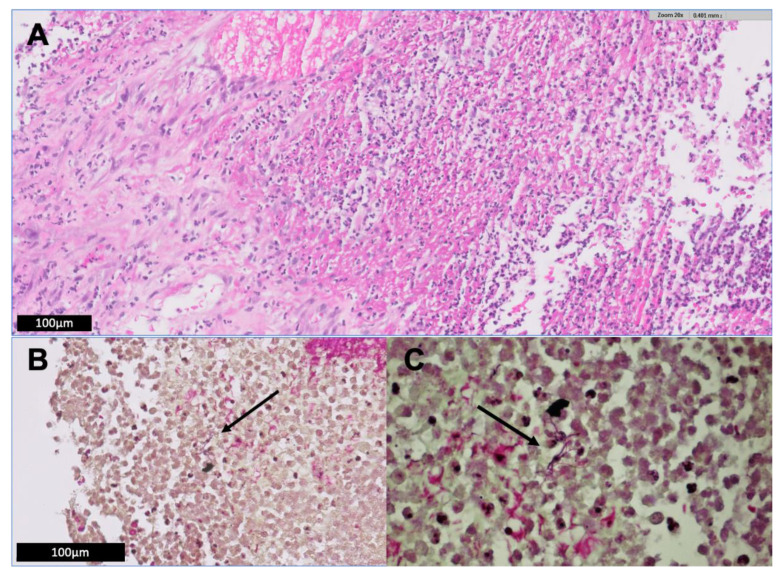
(**A**) Histologic examination of a retroperitoneal biopsy specimen demonstrates a diffuse neutrophilic infiltrate with necrosis and cellular debris. High magnification of a typical area of one of these abscesses with neutrophils, debris and liquefactive necrosis. Organisms are not visible on routine H&E stain. (**B**,**C**) Gram stain shows filamentous bacteria (arrows), consistent with Nocardia species. The organisms are Gram-variable or weakly Gram positive, thin and filamentous, and focally concentrated.

**Table 1 jcm-09-02141-t001:** Previously published cases of *Nocardia paucivorans* infection.

Case n°/Reference	Year **	Age	Sex	Site	Immune Status	Therapy Regimen	Therapy Duration	Outcome
1 [58]	2000	51	M	Sputum	Chronic lung disease	NR	NR	NR
2 [59]	2002	40	F	Brain abscess	NR	NR	NR	NR
3 [60]	2002	63	M	Brain abscess	Low CD4+ count of unknown origin	Cef, Amp, Amk, Mer, Levo, Mino	>6 months	survived
4 [61]	2006	54	M	Sputum, BAL, lung biopsy	Immunocompetent	TMP/SMX, A/clav, Cip	6 months	NR
5 [62]	2006	63	M	Brain abscess	Immunocompetent	Surgical resection, TMP/SMX	3 months	survived
6 [63]	2007	52	M	Lung abscess	NR	NR	NR	died
7 [63]	2007	55	M	Brain abscess	NR	NR	NR	NR
8 [63]	2007	NR	M	Pleural fluid	NR	NR	NR	NR
9 [63]	2007	44	M	Mediastinal lymph node	NR	NR	NR	NR
10 [63]	2007	78	F	Lung abscess	Chronic lung disease	NR	NR	NR
11 [63]	2007	50	F	Sputum	NR	NR	NR	NR
12 [63]	2007	41	M	Brain abscess	Hodgkin’s lymphoma	NR	NR	NR
13 [63]	2007	53	M	Lung abscess	NR	NR	NR	NR
14 [63]	2007	58	F	Brain abscess	Corticosteroid therapy	NR	NR	NR
15 [63]	2007	66	F	Sputum	NR	NR	NR	NR
16 [63]	2007	54	M	Endocarditis, brain abscess,	NR	NR	NR	NR
				skin abscess				
17 [63]	2007	65	F	Sputum	NR	NR	NR	NR
18 [63]	2007	72	F	Lung abscess	Corticosteroid therapy	NR	NR	NR
19 [63]	2007	66	M	Brain abscess	Immunosuppression *	NR	NR	NR
20 [63]	2007	74	M	BAL fluid	NR	NR	NR	NR
21 [63]	2007	NR	M	Sputum	NR	NR	NR	NR
22 [63]	2007	74	M	Skin abscess	NR	NR	NR	NR
23 [63]	2007	57	M	Brain abscess, pneumonia	NR	NR	NR	NR
24 [63]	2007	80	M	Lung, skin	NR	NR	NR	NR
25 [63]	2007	62	M	Skin	Immunocompetent	NR	NR	NR
26 [63]	2007	87	M	Sputum	NR	NR	NR	NR
27 [63]	2007	77	F	BAL fluid	Immunocompetent	NR	NR	NR
28 [63]	2007	50	M	Sputum	Lung cancer	NR	NR	NR
29 [63]	2007	67	M	Brain abscess	Diabetes mellitus	NR	NR	NR
30 [63]	2007	67	M	Sputum	Immunocompetent	NR	NR	NR
31 [63]	2007	47	F	Sputum	NR	NR	NR	NR
32 [63]	2007	66	F	Sputum	Previous lung infection	NR	NR	NR
33 [63]	2007	46	M	Sputum	Immunocompetent	NR	NR	NR
34 [63]	2007	79	M	Pleural fluid	Corticosteroid therapy	NR	NR	NR
35 [63]	2007	76	M	Pleural fluid	Immunocompetent	NR	NR	NR
36 [63]	2007	61	M	Skin abscess	NR	NR	NR	NR
37 [63]	2007	60	M	Blood	Chemotherapy	NR	NR	NR
38 [63]	2007	53	M	BAL fluid and skin lesion	Smoking and other drug abuse, Hepatitis C	Tica/clav, Rox, TMP/SMX	12 months	survived
39 [56]	2014	50	M	M. iliopsoas, brain, lung,	Smoking and other drug abuse	TMP/SMX, Imi, Mox	12 months	survived
				mediastinal lymph nodes				
40 [64]	2015	70	M	Brain abscesses	Multiple Myeloma under chemotherapy	TMP/SMX, Mer, Cip	12 months	survived
41 [65]	2016	50	M	Brain abscesses	Smoking and other drug abuse	TMP/SMX, Imi, Line, Mox	12 months	survived
42 [66]	2016	80	F	Brain and lung abscess	Immunocompetent	Surgical resection, TMP/SMX, Cef	9 months	survived
43 [66]	2016	50	M	Spinal cord, lung	Alcohol abuse	Surgical resection, TMP/SMX, Mer, Cef	9 months	survived
44 [66]	2016	59	M	Brain and lung abscess	Immunocompetent	TMP/SMX, Mer	12 months	survived
45 [67]	2016	54	M	Brain abscess	Immunocompetent	Cef, Met, Mer, Line, Imi, Van, Rif, TMP/SMX	21 weeks	survived
46 [3]	2018	61	M	Brain abscess	Myasthenia gravis, corticosteroid therapy	TMP/SMX, Cef, Met, Line, Levo	11 months	survived
47 [2]	2018	66	F	Renal abscess	Post Lung Transplantation	TMP/SMX, Imi	9–12 months	survived
48 [68]	2018	42	M	Lung empyema	Suspected silicosis	TMP/SMX	NR	survived
49 [4]	2019	52	F	Brain abscess	Immunocompetent	Surgical resection, Cef, TMP/SMX, Levo	13 months	survived
50 [present report]		81	M	Abscess in right adrenal loge	B-Cell Lymphoma, Hypogammaglobulinemia	Surgical resection, A/clav, TMP/SMX, Amk, Cef, Mino, Cip	6 months	survived

**Abbreviations**: Amk = amikacin; Amp = ampicillin; A/clav = amoxicillin + clavulanic acid; BAL = bronchoalveolar lavage; Cef = ceftriaxone; Cip = ciprofloxacin; Imi = imipenem; Levo = levofloxacin; Line = linezolid; Mer = meropenem; Met = metronidazol; Mino = minocycline; Mox = moxifloxacin; NR = not reported; Rif = rifampicin; Rox = roxifloxacin; Tica/clav = ticarcillin + clavulanic acid; TMP/SMX = trimethoprim-sulfamethoxazole; Van = vancomycin; * not further specified; ** Year of publication.

**Table 2 jcm-09-02141-t002:** Previously published cases of isolated abdominal/retroperitoneal nocardiosis.

Case n°/Reference	Year **	Age	Sex	Species	Site	Immune Status	Therapy Regimen	Therapy Duration	Outcome
1 [28]	1976	49	F	NR	Spleen abscess	Immunocompetent	Sulfisoxazole	NR	survived
2 [9]	1981	26	F	*N. asteroides*	Perirenal area *	Post renal transplant	Sulphatriad, fusidic acid	NR	survived
3 [10]	1983	11	F	*N. asteroides*	Liver and kidney abscesses	Immunocompetent	Surgical resection, Amk, TMP/SMX, Sulphadimidine	12 weeks	survived
4 [11]	1986	76	M	*N. asteroides*	Pancreas abscess	Immunocompetent	Surgical resection, A/clav, Amk, Orn, TMP/SMX	NR	survived
5 [41]	1990	19	M	*N. brasiliensis*	Cholecystitis and Peritonitis	AIDS	NR	7 days	died
6 [12]	1991	38	M	*N. asteroides*	Left suprarenal abscess	AIDS	Surgical resection	NR	died
7 [13]	1996	67	M	*N. asteroides*	Abdominal aortic aneurysm	Immunocompetent	TMP/SMX, Amk	NR	survived
8 [14]	2003	42	M	*N. farcinica*	Psoas abscess	Immunocompetent	TMP/SMX	11 months	survived
9 [15]	2004	37	F	NR	Abdominal abscess	M. Crohn	NR	NR	NR
10 [5]	2004	34	M	*N. asteroides*	Left Adrenal abscess	AIDS	Surgical resection, TMP/SMX, Cef	NR	survived
11 [16]	2007	25	F	NR	Bilateral kidney abscesses	Corticosteroids and Azathioprin for SLE	Methylprednisolone	NR	died
12 [17]	2007	44	F	NR	Jejunum	Post liver transplant	Surgical resection, Imi, Van	NR	died
13 [18]	2009	55	M	NR	Kidney abscess	AIDS	Surgical resection	NR	NR
14 [42]	2010	83	F	*N. veterana*	Bowel abscess	Colon cancer	Surgical resection, Cef, Gen, Met, TMP/SMX	NR	survived
15 [19]	2011	32	F	NR	Bilateral abscesses of ovaries and fallopian tubes, omentum	Immunocompetent	Surgical resection, Mino, TMP/SMX, Amk, Line, Cip	6 Months	survived
16 [20]	2012	59	F	*N. farcinica*	Right adrenal compartment	Chronic hepatitis C, post treatment for lymphoma	Amk, Cefu, TMP/SMX	4 months	died
17 [21]	2012	54	F	NR	Colon	Immunocompetent	Surgical resection, antibiotics not specified	NR	survived
18 [22]	2013	30	F	NR	Left adnexal collection *	Immunocompetent	TMP/SMX, Amk	6 months	survived
19 [24]	2016	58	F	*N. farcinica*	Colon, blood	TNFa-antagonist for M. Crohn	Van, Met, Imi, Amk, TMP/SMX	12 months	survived
20 [23]	2017	59	M	*N. farcinica*	Liver abscess	Post liver transplant	TMP/SMX, Amk, Imi	NR	survived
21 [2]	2018	63	F	*N. paucivorans*	Renal abscess	Post lung transplant	Surgical resection, Imi, TMP/SMX	9–12 months	NR
22 [25]	2019	63	M	*N. farcinica*	Liver (hepatitis)	Immunocompetent	TMP/SMX, Amk	4 weeks	died
23 [26]	2019	11	F	*N. farcinica*	Abdominal abscesses	Post renal transplant	TMP/SMX, Mer, Imi, Cila, Line	NR	NR
24 [present report]		81	M	*N. paucivorans*	Right adrenal space	B-Cell Lymphoma, Hypogammaglobulinemia	Surgical resection, A/clav, TMP/SMX, Amk, Cef, Mino, Cip	6 months	survived

**Abbreviations**: AIDS = acquired immunodeficiency syndrome; Amk = amikacin; A/clav = amoxicillin + clavulanic acid; Cef = ceftriaxone; Cefu = cefuroxime; Cila = cilastin; Cip = ciprofloxacin; Gen = gentamycin; Imi = imipenem; Line = linezolid; Mer = meropenem; Mino = minocycline; NR = not reported; Orn = ornidazole; SLE = systemic lupus erythematodes; TNFa = tumor necrosis factor alpha; TMP/SMX = trimethoprim-sulfamethoxazole; Van = vancomycin; * not further specified; ** Year of publication.

**Table 3 jcm-09-02141-t003:** Previously published cases of disseminated nocardiosis with abdominal/retroperitoneal infection.

Case n°/Reference	Year **	Age	Sex	Species	Site	Immune Status	Therapy Regimen	Therapy Duration	Outcome
1 [27]	1975	13	M	*N. asteroides*	Lungs, liver, pancreas, lymph nodes	Immunocompetent	Surgical resection, sulphadiazine, sulfisoxazole	6 months	NR
2 [29]	1997	59	M	*N. otitidiscaviarum*	Thoracoabdominal abscess, lung	HIV positive	Surgical drainage, TMP/SMX, Amk, Cefotaxime	NR	survived
3 [30]	1998	49	F	*N. asteroides*	Left adrenal abscess, spleen, lung	chronic steroids for RA	Surgical resection	NR	survived
4 [34]	2004	32	M	*N. brasiliensis*	Lung, intestines	AIDS	Surgical resection	NR	died
5 [17]	2007	30	M	*N. asteroides*	Lung, pleura, abdomen *	Post kidney transplant, chemotherapy, corticosteroids	NR	NR	survived
6 [31]	2007	67	M	*N. asteroides*	Pancreas, omentum, brain, lungs, thyroid	Chronic steroids for Still’s disease	Mer, Amk, TMP/SMX	NR	died
7 [32]	2009	69	M	*N. farcinica*	Adrenal glands, brain, lung, skin, muscle	Post liver transplant	TMP/SMX	7 months	survived
8 [6]	2010	66	F	*N. farcinica*	Brain, bilateral adrenal abscesses, abdominal lymph nodes	aTNF-therapy for Psoriasis	Van, Amp, Mer, Voriconazole, TMP/SMX, Line	2.5 months	died
9 [33]	2010	61	F	*N. nova*	Lung, skin, kidney, pancreas, brain	Chronic steroids and Azathioprine for ulcerative colitis	TMP/SMX	1 year	survived
10 [35]	2011	42	M	*N. concava*	Lung, liver	Chronic steroids for polychondritis	Sulphadiazine, Van, Imi, Cip, Amk	25 days	died
11 [20]	2012	68	F	*N. farcinica*	Right kidney abscess, brain abscess, lung	Anorexia nervosa	Cef, Erythromycin, TMP/SMX, Amk, Imi, Cip	65 days	died
12 [36]	2014	75	M	*N. farcinica*	Kidney, liver, spleen, lung, brain	Immunocompetent	Mer, Van, Cefepime, Doxycycline, Acilovir	3 days	died
13 [37]	2015	59	F	*N. cerradoensis*	Brain, skin, retroperitoneum, lung	Post renal transplant	Mer, Amk	3 months	survived
14 [40]	2016	37	M	*N. otitidiscaviarum*	Subcutaneous soft tissue, liver, lung	Immunocompetent	Cef, TMP/SMX, Mino	NR	survived
15 [38]	2016	58	M	*N. nova*	Lung, ileum	B-cell Non-Hodgkin lymphoma	TMP/SMX, Imi	11 months	died
16 [39]	2018	12	M	*N. elegans/aobensis/* *africana complex*	Kidney, lung, brain	Immunocompetent	Pip/Taz, Amk, Imi, Cip	6 days	died

**Abbreviations:** AIDS = acquired immunodeficiency syndrome; Amk = amikacin; Cef = ceftriaxone; Cip = ciprofloxacin; HIV = human immunodeficiency virus; Imi = imipenem; Line = linezolid; Mer = meropenem; Mino = minocycline; NR = not reported; Pip/Taz = piperacillin-tazobactam; RA = rheumatoid arthritis; aTNF = anti-tumor necrosis factor therapy; TMP/SMX = trimethoprim-sulfamethoxazole; Van = vancomycin; * not further specified; ** Year of publication.

**Table 4 jcm-09-02141-t004:** *Nocardia* peritonitis in patients with continuous ambulatory peritoneal dialysis (CAPD).

Case n°/Reference	Year **	Age	Sex	Species	Cause of ESRD	Therapy Regimen	Therapy Duration	Outcome	Catheter Removal
1 [43]	1981	70	M	*N. asteroides*	CIN	Ceph, Sulf	6 weeks	survived	No
2 [44]	1990	60	M	NR	PCKD	Ofl, Tob, Ceph, Van, Azt, TMP/SMX	8 weeks	survived	No
3 [45]	1990	58	F	*N. asteroides*	NR	Ceft, Net, Van, TMP/SMX	NR	survived	Yes
4 [46]	1993	38	M	*N. asteroides*	SLE	Ceph, TMP/SMX	4 weeks	died	No
5 [47]	1994	80	M	*N. nova*	NR	TMP/SMX	3 weeks	survived	Yes
6 [48]	2001	32	M	*N. nova*	Type 1 DM	TMP/SMX, Imi	4 months	survived	Yes
7 [49]	2003	68	F	*N. nova*	Unknown	Cefa, Net, Ceft, Amk, Imi, TMP/SMX, Cef	2 weeks	died	No
8 [50]	2005	35	M	*N. asteroides*	CBUS	Van, Ceft, TMP/SMX, Amk, Cefu	19 weeks	survived	No
9 [51]	2008	75	M	NR	Type 2 DM	Van, TMP/SMX	11 weeks	survived	Yes
10 [8]	2008	66	M	*N. asteroides*	Type 2 DM	Cefa, Gen TMP/SMX	12 months	survived	Yes
11 [52]	2011	57	M	*N. asteroides*	Type 2 DM	Van, Ceft, TMP/SMX, Cef, Cip	2 weeks	died	Yes
12 [53]	2016	13	F	*N. asteroides*	DGS	Van, Cip, Ceft, Line	8 months	survived	Yes

**Abbreviations:** Azt = aztreonam; Cef = ceftriaxone; Cefa = cefazolin; Ceft = ceftazidime; Cefu = cefuroxime; Ceph = cephalotin; Cip = ciprofloxacin; CIN = chronic interstitial nephritis; CBUS = congenital bilateral ureteral stenosis; DGS= diffuse global sclerosis; DM = diabetes mellitus; ESRD = end-stage renal disease; Gen = gentamycin; Imi = imipenem; Line = linezolid; Net = netilmycin; NR = not reported; Ofl = ofloxacin; PCKD = polycystic kidney disease; SLE = systemic lupus erythematous; Sulf = sulfisoxazole; TMP/SMX = trimethoprim-sulfamethoxazole; Tob = tobramycin; Van = vancomycin; ** Year of pulication.

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
