# Peer review of "Intra-Abdominal Nocardiosis—Case Report and Review of the Literature"

_jcm, 2020, doi:10.3390/jcm9072141_

Round 1

Reviewer 1 Report

The topic is very interesting and an impressive effort was made in reviewing the literature on the topic. However, the manuscript has fundamental pitfall which should be carefully addressed and corrected by the authors.

Major comments

Title and structure

For publications reporting a specific case and review the literature, it is best to open with the case description and only then to elaborate on the existing literature in the field. Doing so ensures the logic behind embarking on literature search and guides the readers to understand the existing evidence and its limitations. I therefore recommend the authors to reorganize their manuscript, and to open it with the case report (section 4, pages 13-14). In addition, the authors should rephrase the title as follows: Intra-abdominal nocardiosis – a case report and review of the literature".

Abstract

The abstract does not adequately reflect the review's content. The authors are kindly requested to review their entire manuscript and then re-write an abstract. The new abstract should optimally reflect the gap in knowledge which the manuscript aims to address, the methods, the main findings, and the conclusion.   

Introduction

Lines 42-45 deals with an issue which is irrelevant for the introduction section. The introduction should introduce the topic and frame the existing gaps in evidence. The content in these lines suits the conclusions or the concluding parts of the case report. The authors are kindly requested to omit these lines. In additions, the authors should add few sentences regarding the existing evidence on intra-abdominal nocardiosis. These should deal with the lack of evidence regarding treatment options and what could be the difficulties and clinical dilemmas in managing an intra-abdominal Nocardial infection. Appropriate references are also required. This paragraph should be concluded with the aims of this manuscrips (e.g. the aim of this article is to review the existing evidence on treatment and outcomes of intra-abdominal nocardiosis).

Methods

  • The authors included "paucivorans" among their search terms. This highlights the necessity in opening with the case report, as was explained above. Moreover, the authors must explain the aims of their review. It should be emphasized whether it is a review on abdominal nocardiosis or on paucivorans infection? Maybe on both?  
  • Please provide inclusion and exclusion criteria for the literature review.
  • The authors must report how many publications were initially found, how many were excluded, and why.
  • Nocardiosis involving the abdomen is relatively rare, while sole abdominal involvement is even less frequent. Therefore, when the authors define abdominal nocardiosis (lines 51-53), they must address additional systems involvement, i.e. they can state that abdominal nocardiosis is defined as so and so, regardless of the infection's origin and extent of dissemination.

Body of text

  1. Infection due to paucivorans
    • The authors open the report on the literature search with this topic, although it was not mentioned appropriately before. Again, if the manuscript opens with the case report, the readers would be able to understand the rational. Still, the authors should re-design the manuscript, from the title to the conclusions part, so it will address two topics: intra-abdominal nocardiosis and paucivorans infections. In my humble opinion, a good article is an article which addresses a single topic. I leave it for the authors to consider, however if they decide to keep both topics, the entire text should be framed in accordance, including the aims and the rational behind the literature search.
    • The content in lines 76-79 is related to the case report and should not be mentioned in the review section, as it does not contribute to the discussion at this point. Please omit it.
    • The authors should add Table 1 a column with the year of publication, for each published case. The authors should organize the table chronologically – from the earlies to the latest publish case (the present report).
    • This paragraph, which was aimed at addressing the topic of paucivorans infections, provides no in-depth analysis. I expect the authors to report what can be taught from the literature. For example, what was the most prevalent drug/regimen? Did it affect the outcome? etc. Otherwise, this review contributes nothing to the body of knowledge.
    • There are significant limitations concerning the quality of reported cases (regimens and outcomes were reported in minority of the publications). The authors should explicitly mention these limitations in the text.

  1. Abdominal/retroperitoneal infection due to Nocardia species
    • This section is the core of the manuscript; however the analytic content of this paragraph is quite little. The information presented in the tables must be translated into original insights of the authors. The authors must re-write this section and address the main findings arise from the literature review.
    • The authors should add Tables 2a and 2b a column with the year of publication, for each published case. The authors should organize the table chronologically – from the earlies to the latest publish case.

  1. Peritoneal dialysis-related Nocardia peritonitis
    • The authors should have mentioned this topic in the introduction section. It should be included among the aims of this article. Additionally, peritoneal dialysis-related Nocardia peritonitis should have been defined earlier. The inclusion/exclusion criteria for the literature in the field were also to be mentioned. The authors must fundamentally correct this along the entire manuscript.
    • The authors should add Table 3 a column with the year of publication, for each published case. The authors should organize the table chronologically – from the earlies to the latest publish case.

  1. Route of Nocardia acquisition and dissemination

As the reviewed publications reported no information on the presumed route of acquisition, I find this section irrelevant for the review. The authors should be aware of their aims, as it is not a narrative review on nocardiosis in general.

  1. Nocardia diagnosis

The topic discussed in this section is beyond the scope of the current review. The authors should consider omitting it.

  1. Treatment of Nocardia paucivorans

This part should be integrated into section 3.1. However, the content should be rephrased to address the finding of the literature search (see above).

Case report

  • The case should be reported at the beginning of the manuscript. The methods and literature findings should be provided only consequently.
  • Please specify whether the patient had an immunosuppression (corticosteroid therapy, chemotherapy) during the six months prior the diagnosis with nocardiosis.
  • As for species level identification, please provide information on the technique (was it 16S rRNA sequencing?), the PCR primers' manufacturer etc.
  • Please elaborate on how antimicrobial susceptibility testing was performed (broth microdilution? E-test? Antimicrobial disk diffusion?).
  • Please address the extent of the infection. Was a brain imaging performed (as recommended for all patients diagnosed with nocardiosis)? Were the lungs involved?
  • Please specify on whether you presumed a route of acquisition and infection?
  • Was an abdominal imaging conducted at 6-months follow-up?

Conclusions

  • Please rephrase the conclusions section as a paragraph, as appropriate in the medical literature.
  • The conclusions should be organized in an order that addresses the article's aims. Please make the necessary adaptations; in order to keep the manuscript's rational from the beginning to its end.
  • I am afraid that none of the mentioned conclusions are in fact concluded from the literature search. The authors should discern between assumptions or individual lessons from the specific case reported and conclusions arising from the reviewed literature. The authors are kindly requested to review the main findings from the literature and discuss these in details within the main body of the manuscript. When done so, the main conclusions should be summarized in this section.  

Minor comments

  • The authors must review their manuscript and be punctilious about writing Nocardia using a capital N and in italic type, as is common it the literature.
  • The authors should rephrase lines 40-42 to open with the case report: "here we report the second case of… and provide a comprehensive review on…".
  • Nocardiosis primarily occurs in individuals subjected to immunosuppression. Therefore, lines 56-59 should be reorganized so that organ transplantation, malignancies and HIV infection will appear before diabetes.
  • When data is provided in proportions, the reader can grasp the concepts beyond the figures. Therefore, in lines 60-79, 93-94 and so on, the authors should add the percentage whenever a number is mentioned (e.g. 33 cases, 67%).  
  • The figures in this paragraph do not total in 49 (e.g. 33+18>49). Please rephrase it and make it clear for the readers what sub-populations are you talking about ("of the remaining..." etc.).   
  • In line 69, the authors can phrase: microbiologically proved lung involvement instead of the detailed description added in brackets.
  • In line 74, the case report of this manuscript is mentioned in section 4 and not in section 3. As mentioned above, it is recommended to open the manuscript with the case report; accordingly the authors will be able to mention it at any further point.

Reviewer 2 Report

This article is an extensive literature review of published cases of intra-abdominal Nocardia infections, and cases of Nocardia paucivorans infections, including one new case presentation of intra-abdominal infection caused by this pathogen. The manuscript is well-written and the figures appropriate.

My suggestions:

1.Introduction: Please start with a brief presentation of Nocardia species (one sentence)

2.Consider re-arrangement of sections: firstly the case presentation, and then the literature review-discussion.

3. In the discussion section consider adding a paragraph about the association of the type of immunosuppression with Nocardia infections. Do all immunocompomised patients have the same risk? Does the clinical presentation and/or severity differ? You may add this information inside the paragraph 'Route of Nocardia acquisition and dissemination' or you may write a different paragraph.

4.Tables: In the column of sex you use 3 abbreviations: M, F, and W. Does W stand for woman? Please use only F.

5.Line 96: Inside the parentheses should it be Table 2a-2b?

Round 2

Reviewer 1 Report

The MS in its current version has been tremendously improved. 

The topic is interesting. However, the abundance of issues discussed diminishes the MS thematic coherence. Usually, a good study deals with a single question. Here, the authors systematically reviewed both abdominal nocardiosis and N. paucivorans infections. It is reasonable as their case report dealt with N. paucivorans abdominal infection. However, addressing general issues regarding nocardiosis (i.e. section 4.1. clinical characteristics of nocardiosis) seems beyond the scope of the current study. As a case report and review of the literature on abdominal nocardiosis, it should not encompass general aspects of nocardiosis. The scientific value of these short paragraphs is low, and much more could have been written about each of these topics, if required. In my humble opinion, section 4.1. should be omitted, and instead the authors are encouraged to enrich their discussion on abdominal nocardiosis (risk factors, diagnosis, and treatment considerations). 

Author Response

Dear Dr Yan, Dear Editor

Thank you for the opportunity to submit a revised version of our work and for the insightful comments by the reviewers that have helped us improve our manuscript.

Below please find the detailed answers to the reviewers’ comments. We have highlighted all changes in the manuscript in track mode.

Thank you for re-considering our manuscript for publication in JCM.

I will serve as correspondent.

Sincerely, Philip E. Tarr, MD